# Intracranial Response Rate in Patients with Breast Cancer Brain Metastases after Systemic Therapy

**DOI:** 10.3390/cancers14040965

**Published:** 2022-02-15

**Authors:** Anna Niwinska, Katarzyna Pogoda, Agnieszka Jagiello-Gruszfeld, Renata Duchnowska

**Affiliations:** 1Department of Breast Cancer and Reconstructive Surgery, Maria Sklodowska-Curie National Research Institute of Oncology, Roentgena 5 Str, 02-781 Warsaw, Poland; anna.niwinska@pib-nio.pl (A.N.); agnieszka.jagiello-gruszfeld@pib-nio.pl (A.J.-G.); 2Department of Oncology, Military Institute of Medicine, Szaserów 128 Str, 04-141 Warsaw, Poland; rduchnowska@wim.mil.pl

**Keywords:** brain metastases, breast cancer, CNS, intracranial response, targeted therapy

## Abstract

**Simple Summary:**

For many years, patients with breast cancer and brain metastases were excluded from participation in clinical trials. It was believed that anticancer drugs could not cross the blood–brain barrier. However, recent evidence strongly suggests that some drugs can act against brain metastases, with the greatest intracranial response rate reported in the case of capecitabine, neratinib plus capecitabine, trastuzumab deruxtecan and tucatinib plus trastuzumab and capecitabine. In this article, we discuss the achievements in systemic therapy of breast cancer patients with brain metastases. We stress on the newest clinical trial results which indicate tremendous progress in HER2-positive breast cancer. On the other hand, in patients with triple-negative breast cancer or hormone-receptor-positive brain metastases, much fewer compounds were discovered. Based on the presented results, patients with active brain metastases should be routinely included in clinical trials with novel agents.

**Abstract:**

Brain metastases are detected in 5% of patients with breast cancer at diagnosis. The rate of brain metastases is higher in HER2-positive and triple-negative breast cancer patients (TNBC). In patients with metastatic breast cancer, the risk of brain metastases is much higher, with up to 50% of the patients having two aggressive biological breast cancer subtypes. The prognosis for such patients is poor. Until recently, little was known about the response to systemic therapy in brain metastases. The number of trials dedicated to breast cancer with brain metastases was scarce. Our review summarizes the current knowledge on this topic including very significant results of clinical trials which have been presented very recently. We focus on the intracranial response rate of modern drugs, including new antibody–drug conjugates, HER2- targeted tyrosine kinase inhibitors and other targeted therapies. We highlight the most effective and promising drugs. On the other hand, we also suggest that further efforts are needed to improve the prognosis, especially patients with TNBC and brain metastases. The information contained in this article can help oncologists make treatment-related decisions.

## 1. Introduction

Breast cancer has come to be recognized as the second most common solid tumor that metastasizes to the central nervous system (CNS) [1]. This seems to be the consequence of the increased detection of metastatic disease through advanced imaging techniques and thus improved survival rates of patients with metastatic breast cancer.

At initial diagnosis, brain metastases are detected in 5% of patients with breast cancer. The incidence of brain metastases varies amounting to 1% in luminal A, 2% in luminal B, 4% in human epidermal growth factor receptor 2 (HER2)-positive and 6% in triple-negative (TNBC) breast cancer. In patients with metastatic breast cancer, the risk of brain metastases is much higher. In HER2-positive breast cancer and TNBC, the rates of brain metastases are 11–49% and 26–46%, respectively, while in patients with metastatic luminal A and B cancer, the risk of brain metastases are 8–15% and 11%, respectively [2,3,4,5,6,7,8].

Current therapeutic options for patients with breast cancer brain metastases include surgical resection, stereotactic radiosurgery (SRS), whole-brain radiation therapy (WBRT), and systemic therapy (i.e., chemotherapy and targeted therapy) [9]. Until recently, the efficacy of the systemic therapy in patients with brain metastases had not been studied in clinical trials because it was generally believed that the blood–brain barrier (BBB), formed by a tight junction of endothelial cells, a thick basement membrane of pericytes, and astrocytic end-foot processes [10], was not permeable for almost all drugs. Patients with brain metastases were, consequently, excluded from randomized clinical trials assessing the efficacy of new systemic agents. Among the 1474 clinical trials concerned with metastatic breast cancer carried out, only 16 (1%) concerned CNS metastases from breast cancer [11]. Meanwhile, basic research has shown that the blood–tumor barrier (BTB) is more permeable to some agents than the BBB, especially in the case of macrometastases with neovascularization [12,13,14]. Moreover, retrospective and prospective observational clinical studies seem to indicate the efficacy of some systemic drugs in patients with multiple and single brain metastases and even in solitary brain metastases without dissemination to other organs [15,16,17]. There are suggestions that BBB can be disrupted by neovascularization during progression and become BTB. On the other hand, micrometastases and meningeal carcinomatosis have less abnormal vessels which prevent access to systemic therapies [9]. This observation could force other strategy to deliver effective solution. Additionally, there can be differences in brain metastases location in breast cancer patients compared to other primary tumors. Brain metastases from breast cancer were more frequently located in cerebellum in retrospective study which could have occurred due to increased cortico-junctional surface and better perfusion [18,19].

Until recently, relatively little was known of the intracranial objective response rate to systemic therapy. However, research in this field is given ever more attention. According to ClinicalTrial.gov, more than 100 trials on breast cancer brain metastases are now in progress, including studies focusing on new agents (cyclin-dependent kinase (CDK) inhibitors, poly (ADP-ribose) polymerase (PARP) inhibitors, phosphoinositide 3-kinase (PI3K) inhibitors, new generation tyrosine kinase inhibitors (TKIs), anti-HER2 conjugates, immune drugs) with the evaluation of the objective intracranial response rate as one of the study endpoints [9].

## 2. Aim

The aim of this overview was to analyze the objective intracranial response rate (iORR) and the clinical benefit rate (iCBR; complete response (CR) + partial response (PR) + stable disease (SD)) of known and new systemic agents used in breast cancer patients with brain metastases including the most recent clinical trial results.

## 3. HER2-Positive Breast Cancer

The group of patients with HER2-positive breast cancer and brain metastases is best represented with regard to the number of studies assessing the intracranial response rate after cytotoxic drugs and targeted therapy. A meta-analysis of 97 studies by Erickson et al. showed that HER2-targeted therapy was associated with improved overall survival (OS) (hazard ratio (HR) 0.47; 95% confidence interval (CI) 0.39–0.56) in patients with HER2-positive breast cancer and intracranial metastases compared to the nontargeted therapy group. The iORR was 22% (95% CI, 14–30%), intracranial disease control rate (iDCR) was 62% (95% CI, 55–69%), and intracranial complete response rate was (iCRR) 0% (95% CI, 0–0.01%), with grade ≥3 adverse event rate of 26% (95% CI, 11–45%). However, anti-HER2 therapy did not have a significant effect on the progression-free survival (PFS) (HR 0.52; 95% CI, 0.27–1.02) [20].

### 3.1. Monoclonal Antibodies

***Trastuzumab*** has been proved to be a monoclonal antibody that is very effective in controlling extracranial disease and prolonging the survival of patients with disseminated extracranial and intracranial disease [15], however its permeability through the BBB is limited [3]. Biodistribution of trastuzumab into the brain is a little better in patients after WBRT or in those with leptomeningeal metastases but it seems that the efficacy of this monoclonal antibody in breast cancer patients is rather the result of extracranial disease control [4].

***Pertuzumab*** is a humanized monoclonal antibody preventing HER2/HER3 dimerization and resulting in a double blockage when associated with trastuzumab. In the CLEOPATRA randomized trial with trastuzumab plus docetaxel vs. trastuzumab plus docetaxel plus pertuzumab, a major impact on the OS in the pertuzumab arm was shown (56.5 vs. 40 months), but the trial did not include patients with brain metastases [21]. During the observation, the rate of newly detected CNS metastases in both arms was comparable (13.7% and 12.6%, *p* = not significant (NS)) but the onset of CNS metastases was delayed in the pertuzumab group (15 vs. 11.9 months, *p* = 0.005) probably due to better control of extracranial disease by pertuzumab [22]. In the phase II single arm PATRICIA trial, 40 patients with progressing brain metastases after previous radiation therapy received pertuzumab (a loading of dose 840 mg, then 420 mg every 3 weeks) plus a high dose of trastuzumab (6 mg/kg weekly) [23,24,25]. The iORR was 11% and the 6-month iCBR was 51%.

### 3.2. Antibody-Drug Conjugates

***Trastusumab emtansine (T-DM1)*** is a conjugate of trastuzumab and emtansine. Retrospective studies confirm the intracranial efficacy of T-DM1 with an iORR of 24–44% and a CBR of 55–70% [24,26,27,28,29,30]. The analysis of 10 patients revealed a 30% iORR rate during a period of 5 months [26]. In another group of 39 patients, the rate of iORR was 44% and the rate of iCBR was 59% [27]. In an Italian retrospective study, the iORR in 53 patients with brain metastases was 24.5% and the CBR was observed in 55% of the patients for a period of 7 months [28]. The EMILIA randomized phase III trial compared T-DM1 vs. lapatinib plus capecitabine in 991 HER2-positive breast cancer patients previously treated with trastuzumab [31,32]. T-DM1 was shown to prolong the OS as compared with lapatinib plus capecitabine. In a retrospective, exploratory analysis of 90 patients with brain metastases included in the EMILIA trial [33], it was revealed that patients with asymptomatic brain metastases treated with T-DM1 lived longer than those treated with lapatinib plus capecitabine (26.8 months vs. 12.9 months, respectively, HR = 0.38; *p* = 0.008), but the time to progression in the brain did not differ in the two arms (5.9 vs. 5.7 months, respectively). The KAMILLA trial was a single-arm phase III b study evaluating the safety of T-DM1 in HER2-positive locally advanced or metastatic breast cancer previously treated with HER2-targeted therapy and chemotherapy [29]. Among the 2220 patients enrolled into the study, 398 were patients with stable or occult brain metastases. The intracranial response rate was assessed in 126 patients [30]. The PR according to RECIST 1.1 (Response Criteria in Solid Tumors 1.1) criteria was achieved in 42.9% of patients and the CBR in 67%. The median PFS and OS were 5.5 and 18.9 months, respectively.

***Trastuzumab-Deruxtecan (T-DXd)*** is an anti-HER2 monoclonal antibody linked to the topoisomerase inhibitor via a cleavable linker. T-DXd binds HER2 on tumor cells, then is internalized. After that the linker is cleaved within the cell by lysosomal enzymes and released DXd inhibits topoisomerase I–DNA complexes. This process leads to tumor cell apoptosis [34]. T-DXd was approved for use in the United States and the European Union in unresectable or metastatic HER2-positive breast cancer patients who have received two or more prior anti-HER2 based regimens in the metastatic setting based on the DESTINY–Breast 01 trial [35]. In that trial, 261 patients received T-DXd and 263 received T-DM1. A subgroup analysis of 24 patients with brain metastases revealed an iORR of 58.3% [36]. During the San Antonio Breast Cancer Symposium 2021, the results of a subanalysis of the phase III study DESTINY–Breast 03 [37] was presented. T-DXd was compared with T-DM1 in 524 unresected or metastatic breast cancer previously treated with trastuzumab and taxanes. In that trial, patients with clinically stable, treated brain metastases could be administered the treatment at least 2 weeks after WBRT. In total, 62 patients with brain metastases in the T-DXd arm and 52 patients in the T-DM1 arm were included and in 43 and 39 of them, respectively, brain metastases were detected at baseline. In all patients with brain metastases computed tomography or magnetic resonance was performed at baseline and monitored throughout the study. The median follow-up of that brain metastases group was 15.9 months. The 12-month PFS rate in patients treated with T-DXd was 75.8% and in those treated with T-DM1 was 34%, HR = 0.28 (*p* < 0.001). In patients with brain metastases at baseline, the median PFS after T-DXd and T-DM1 was 15 months and 3 months, respectively. The 12-months PFS rate was 72% and 21%, respectively (HR = 0.25). The iORR assessed with RECIST 1.1 criteria in the group with T-DXd was 63.9% (CR 27.8%; PR 36.1%) and in the group with T-DM1 it was 33.4% (CR 2.8%, PR 30.6%). The iCR in the T-DXd and the T-DM1 group were 27.8% and 2.8%, respectively. The PR in the groups was 36.1% and 30.6%, respectively. The results confirm that the T-DXd treatment is associated with a substantial intrathecal response and reduction in CNS parenchymal disease. T-DXd demonstrated a manageable and tolerable safety profile [34]. The second SABCS 2021 abstract described DEBBRAH phase II trial [38]. Here, 39 patients with HER2-positive or HER2-low expression breast cancer and brain or leptomeningeal metastases were treated with T-DXd. In total, 17 patients had luminal B biological subtype. The iORR was achieved in 44% and iCRB in 55% of patients.

### 3.3. HER2- Targeted Tyrosine Kinase Inhibitors (TKIs)

***Lapatinib*** is an oral, small dual TKI that binds HER2 and HER1 receptors. In a phase II study of lapatinib as a single agent used in heavily pretreated patients, the volumetric CNS iORR was 6% [39]. In another phase II study, lapatinib plus capecitabine was evaluated in 50 patients and an iORR of 20% was shown [39]. In the study by Cetin [40], 85 out of 203 patients with brain metastases were treated with lapatinib plus capecitabine and the iORR of 27% of them was achieved. Sutherland treated 34 patients with brain metastases with the same regimen and in 21% of them an iORR was noted [41]. In a systematic review and pooled analysis, the efficacy of 799 patients treated with lapatinib plus capecitabine was analyzed [42]. The iORR was 29.2%, the PFS was 4.1 months and the OS was 12.2 months. In the LANDSCAPE trial, the intracranial response rate after lapatinib plus capecytabine in previously untreated patients with brain metastases was assessed. The volumetric iORR was 66%, suggesting that such a combined therapy, instead of radiotherapy, could be a feasible first-line treatment in HER2-positive breast cancer brain metastases [43]. Lapatinib was investigated in association with temozolomide (TMZ) in heavily pretreated patients with HER2-positive breast cancer and brain metastases, in phase I LAPTEM trial. Stabilization of the disease was achieved in 66.7% out of the 15 treated patients with a median PFS of 2.6 months and a median OS of 10.9 months [44]. A phase II randomized trial compared the iORR in patients treated with SRS and lapatinib plus capecitabine and patients treated with SRS and lapatinib plus topotecan. Only 22 out of the 110 planned patients were included in the study due to high toxicity in the experimental arm. The iORR defined as a >= 50% volumetric reduction was 38% and 0%, respectively [45].

***Neratinib*** is an oral, irreversible pan-inhibitor of HER2-TKI family that binds HER1, HER2 and HER4 receptors. In the Translational Breast Cancer Research Consortium (TBCRC) 022 phase II trial, the efficacy of neratinib was assessed in patients with pretreated, progressing brain metastases. Four different groups of patients were included: with neratinib monotherapy (cohort 1), with neratinib after surgical resection (cohort 2), with neratinib plus capecitabine in patients without (cohort 3A) or with previous lapatinib therapy (cohort 3B), and with neratinib plus ado-trastuzumab emtansine (cohort 4). In 40 patients treated with neratinib monotherapy, 78% were after WBRT. In that group, the iORR rate was only 8%. In patients treated with neratinib plus capecitabine without previous treatment with lapatinib the iORR was 49% (95% CI: 32–66) and in patients treated previously with lapatinib iORR was 33% (95% CI: 10–65). Time to progression was 5.5 and 3 months and OS was 13 and 15 months, respectively [46]. In the phase III NALA randomized trial, neratinib plus capecitabine was compared to lapatinib plus capecitabine in heavily pretreated patients (307 vs. 314, respectively). Patients with symptomatic brain metastases were excluded from the trial but patients with occult brain metastases were included. One-year PFS rate after neratinib plus capecitabine vs. lapatinib plus capecitabine was 37.8% and 14.8%, respectively. OS did not differ significantly in both groups. The time to intervention for brain progression was longer in neratinib group (overall cumulative incidence 22.8% in neratinib group and 29.2% in lapatinib group, *p* = 0.04) [47]. Grade 3 diarrhea in the group with neratinib and lapatinib was observed in 24% and 12%, respectively. Based on these results in 2019, NCCN decided to indicate both capecitabine plus lapatinib and capecitabine plus neratinib as therapeutic options in HER2-positive breast cancer patients with brain metastases. In another phase III NEfERT trial, neratinib plus paclitaxel was compared with trastuzumab plus paclitaxel in patients with advanced breast cancer. In the neratinib arm longer time to new brain metastases and lower rate to brain metastases’ progression (8.3% vs.17.3%, respectively, *p* = 0.045) was observed [48].

***Afatinib*** is an oral, irreversible HER1 and HER2 TKI. In the LUX-Breast 3 phase II study, 121 HER2-positive breast cancer patients with progressive or recurrent brain metastases after trastuzumab and/or lapatinib treatment, were randomized into one of 3 cohorts: with SRS and afatinib alone, with SRS and afatinib plus vinorelbine or with SRS and treatment of the physician’s choice [49]. The iCBR defined as no progression in CNS invasion after 3 months were 30%, 34% and 41.9%, respectively, and the toxicity profiles were worse in the afatinib-containing regimens.

***Tucatinib*** is an oral, reversible HER2 and HER1 TKI whose active metabolites can cross the BBB. In a phase I study, tucatinib plus capecitabine and trastuzumab were tested in 12 patients with brain metastases. In (5) 42% of patients, the iORR was assessed with RECIST 1.1 [50]. In a phase II randomized HER2CLIMB study, tucatinib plus capecitabine and trastuzumab was compared with placebo plus capecitabine and trastuzumab in patients previously treated with trastuzumab, pertuzumab, and T-DM1. In this trial, 612 patients with stable brain metastases were included [51]. In the tucatinib arm, improved PFS and OS values were observed. For patients with brain metastases, the 1-year PFS in the tucatinib-combination group was 29.4%, while in the placebo-combination group it was 0%, *p* < 0.001 [49]. The updated analysis of 291 patients with brain metastases participating in a HER2CLIMB study [52] revealed that the risk of intracranial progression or death was reduced by 61% in the tucatinib arm. The median PFS in the brain was 9.9 months in the tucatinib arm vs. 4.2 months in the control arm. The iORR in the tucatinib arm and the control arm was 47.3% and 20%, respectively (*p* = 0.03). The risk of death was reduced by 40% in the tucatinib arm. The median OS was 21.6 months and 12.5 months in favor of tucatinib. The authors concluded that the addition of tucatinib to trastuzumab and capecitabine doubled the iORR, reduced the risk of intracranial progression or death by two thirds, and reduced the risk of death by nearly a half. That regimen was the first to demonstrate an improved anticancer activity in the brain of patients with HER2-positive breast cancer in a randomized trial. The side effects after tucatinib (rash and diarrhea) were reduced compared with neratinib and lapatinib. Tucatinib, in combination with trastuzumab and capecitabine, has received the US Food and Drug Administration (FDA) and the European Medicines Agency (EMA) approval to be administered to women, with previously treated advanced HER2-positive breast cancer, with or without brain metastases.

***Pyrotinib*** is an oral, irreversible TKI targeting HER1, HER2 and HER4. In the phase III randomized PHOEBE trial [53], 266 patients with HER2-positive metastatic breast cancer previously treated with trastuzumab, taxanes and/or anthracyclines were allocated to a pyrotinib plus capecitabine or a lapatinib plus capecitabine group. The median PFS in the pyrotinib and the lapatinib group was 12.5 and 6.8 months, respectively (HR = 0.48, *p* < 0.0001). The median OS in the pyrotinib group was not reached and in the lapatinib group was 26.9 months (HR = 0.69, *p* = 0.02). Based on this trial, in 2020, China granted full approval of pyrotinib in combination with capecitabine as a second-line standard-of-care treatment for HER2-positive metastatic breast cancer. In the PHOEBE trial, only 11% of patients presented brain metastases. In contrast, the PERMEATE study [54] assessed 78 patients with brain metastases who were divided into two cohorts: a cohort with no prior CNS radiotherapy (A) and a cohort with patients after radiotherapy (B). Both groups were treated with pyrotinib plus capecitabine. The intracranial ORR in cohorts A and B was 75% and 42%, respectively. The CNS PFS in the groups was 11.3 (A) and 5.6 (B) months. This study confirmed the CNS activity of pyrotinib.

***Epertinib*** is a reversible inhibitor of HER1, HER2, HER3 and HER4. In a I/II phase trial, 45 patients with breast cancer, including 5 patients with brain metastases, were assigned to one of 3 groups: with epertinib plus trastuzumab (arm A), with trastuzumab plus vinorelbine (arm B), and with trastuzumab plus capecitabine (arm C). The PR in brain metastases was achieved by one of 2 patients in arm C and SD (≥ 6 months) was seen in 2 of 3 patients with brain metastases in arm A [55].

### 3.4. Other Targeted Therapy

***Cabozantinib*** is a small, multiple TKI that inhibits mesenchymal–epithelial transition factor (MET) and vascular endothelial growth factor (VEGF). In contrast to other tumors such as non-small cell lung cancer and renal cell carcinoma, the efficacy of cabozantinib in heavily pretreated breast cancer patients with brain metastases is modest (iORR per RECIST 5.6%) [56].

***Bevacizumab***, a humanized anti-VEGF monoclonal antibody normalizes peritumoral vessels and enhances drug delivery to brain tumors. It was tested in two studies in patients with brain metastases in combination with other cytotoxic agents. In one study, bevacizumab, carboplatin and trastuzumab (in HER2-positive breast cancer patients) were assessed in 39 patients with progressive brain metastases. The iORR by composite criteria and RECIST was 63% [57]. In the second study [58], patients with WBRT-refractory brain metastases received bevacizumab followed by etoposide and cisplatin. The iORR according to the volumetric criteria was 77% and by RECIST—54.3%.

## 4. Luminal Breast Cancer

In estrogen receptor (ER)-positive metastatic breast cancer and brain metastases, some responses to tamoxifen [59], aromatase inhibitors [60] and fulvestrant [61] were reported in case-series studies.

***Everolimus*** was investigated in combination with vinorelbine and trastuzumab in a study of 32 patients with progressive HER2-positive brain metastases, but the intracranial response rate was only 4% [62].

***Abemaciclib*** is a selective CDK 4/6 inhibitor used in patients with hormone receptor (HR) positive HER2-negative metastatic breast cancer along with an endocrine therapy. Abemaciclib has a higher BBB penetration compared with other CDK4/6 inhibitors. The efficacy of drug in patients with brain metastases was assessed in a nonrandomized phase II study. In total, 58 HR-positive HER2-negative breast cancer patients with and 27 HR-positive HER2-positive patients with brain metastases were treated with abemaciclib. The iORR was 5.2% and 0%, respectively. The iCBR was 24% and 11%. The median OS was 12.5 and 10 months, respectively. The study failed to document the efficacy of abemaciclib in the groups of heavily pretreated patients with brain metastases [63].

## 5. Triple-Negative Breast Cancer

***PARP inhibitors***: iniparib, olaparib, talazoparib, and veliparib were evaluated in metastatic TNBC. The phase II TBCRC 018 trial, including 34 patients treated with iniparib plus irinotecan, showed modest benefit of a 12% iORR. The median PFS was 2.1 months and the median OS was 7.8 months [64]. In the phase III EMBRACA trial assessing patients with BRCA-mutated advanced HER2-negative breast cancer, 15% of patients in the talazoparib arm had brain metastases at baseline [65]. In that subgroup, the PFS was better than in patients without CNS metastases, but the assessment of the intracranial response was not performed.

***Ang1005*** is a novel taxane agent, it consists of 3 paclitaxel molecules covalently linked to Angiopep-2 which help to cross the BBB. In a Phase II study, ANG 1005 was tested in 72 breast cancer patients with CNS metastases. The iORR was 15% and the iCBR was reported in 77% of patients [66].

***Etirinotecan pegol*** is a long-lasting topoisomerase-1 inhibitor. The phase III ATTAIN trial investigated etirinotecan pegol compared to the standard care in 178 patients with TNBC with stable brain metastases [67]. The patients had been treated with anthracycline, taxane and capecitabine previously. There was no difference in the OS between the two arms (7.8 and 7.5 months) as well as in PFS brain metastases (3.9 and 3.3 months).

***Eribulin mesylate*** was evaluated in a prospective study of 118 heavily pretreated breast cancer patients. The iORR was reported in 16% (12) of patients with brain metastases, with a PFS of 5.2 months [68].

***Sacituzumab govitecan (SG)*** is a conjugate of an antibody and active metabolite of irinotecan. In a phase III ASCENT study, 529 patients with TNBC received SG or treatment of the physician’ choice (TPC). 61(12%) of patients had stable brain metastases. In that group, the PFS in SG and TPC was 2.8 and 1.6 months, the median OS was 6.8 and 7.5 months. The iORR was 3% and 0% and the iCBR was 9.4% and 3.4%, respectively [69].

***Immune therapy*** was evaluated in TNBC randomized clinical trials but not exclusively in patients with brain metastases. In the phase III Impassion−130 trial, 902 patients with metastatic TNBC were randomized to *atezolizumab* (an antibody that targets the programmed death (PD) ligand 1) with nab-paclitaxel or placebo with nab-paclitaxel. Only 7% of patients in each arm had brain metastases, but, unlike in the whole group, no significant benefit from atezolizumab was observed in patients with brain metastases [70]. Based on TriNetX real-world and in-house database [71], 3449 TNBC patients treated with immune checkpoint inhibitors and 3461 patients nontreated with these agents were compared with regard to the OS. The median OS in each group was 23.9 vs. 11.6 months, respectively (HR = 0.87). However, there is lack of evidence for the effectiveness of immune checkpoint inhibitors for treating brain metastases from TNBC, since approval for their use in this context was obtained relatively recently [68]. The immune-oncology therapy using atezolizumab (NCT03483012), nivolumab (NCT03807765), and pembrolizumab (NCT03449238) with SRS are under investigation. The first results from nivolumab trial showed that immune therapy in combination with SRS in 12 patients was safe and median intracranial control was 6.2 months [72]. The trials assessing the combined strategies of immune therapy with vaccine are ongoing (e.g., NCT04348747).

The results of clinical trials with breast cancer patients and brain metastases are summarized in Table 1. 

## 6. Old Cytotoxic Drugs

Some retrospective and a few prospective studies investigated the response to various cytotoxic drugs in brain metastases from breast cancer. The ORR in patients treated with cyclophosphamide, fluorouracil, methotrexate, epirubicine (CMF, FEC), cisplatin, and etoposide was 38–59% and median OS was 7–13 months [73,74]. The efficacy of capecitabine as a single agent was shown in small series of breast cancer patients with recurrent brain metastases [75,76,77,78]. In another study, the combination of capecitabine with temozolomid demonstrated the ORR of 18% in patients who recurred after WBRT [79]. The evidence of the vinorelbine efficacy in the treatment of breast cancer brain metastases is also poor. There was no response after temozolomide combined with vinorelbine in patients with brain metastases [80]. However, in a prospective study assessing the chemotherapy with cisplatin and etoposide for patients with brain metastases from breast cancer, non-small cell lung cancer or malignant melanoma, the iORR was 38% [81].

## 7. Conclusions

For many years, patients with breast cancer and brain metastases were excluded from participation in clinical trials because it was believed that anticancer drugs could not cross the BBB. In those days, the authors of American Society of Clinical Oncology (ASCO) recommendations did not consider instituting a systemic therapy in patients whose systemic disease was not progressive at a time of brain metastasis diagnosis. No information was posted in relation to a systemic therapy after the detection of brain lesions without metastases to other organs [82]. However, ample evidence strongly suggests that some drugs can act against brain metastases, with the greatest intracranial response rate reported in the case of capecitabine, neratinib plus capecitabine, T-DXd, and tucatinib plus trastuzumab and capecitabine. Based on the new European Association of Neuro-Oncology (EANO)–European Society for Medical Oncology (ESMO) clinical practice guidelines, a systemic therapy can play an important role in the control of brain metastases from breast cancer and it should be considered for most patients with initial brain metastases, not only after an intracranial recurrence. In asymptomatic brain metastases, a systemic treatment should be considered to delay WBRT [83].

Patients with HER2-positive breast cancer and brain metastases have now several treatment options, but in patients with TNBC or hormone receptor positive brain metastases much fewer compounds can be proposed (Table 2). Improvements in the systemic therapy for TNBC and brain metastases are urgently needed because of the worst outcome in this group of patients. Based on the presented results, patients with active brain metastases should be routinely included in clinical trials of current and novel agents.

## Figures and Tables

**Table 1 cancers-14-00965-t001:** Intracranial response rate based on the results of available clinical trials including patients with breast cancer and brain metastases.

Main Agent	Type of Study	Number of Patients with Brain Metastases	Scheme of Treatment	Intracranial Response Rate *	Median PFS/OS (Months)
	**HER2-positive breast cancer brain metastases**				
Pertuzumab	PATRICIA phase II trial [23]	39	Pertuzumab + high dose trastuzumab	iORR 11% iCBR 68% 4 mo. iCBR 51%6 mo.	-
Trastuzumab-emtansine (T-DM1)	Bartsch [26] retrospective	10	T-DM1	iORR 30% iCBR 50%	PFS 5
	Jacot [27] retrospective	39	T-DM1	iORR 44% iORR 59%	PFS 6
	Fabi 2018 [28] retrospective	70	T-DM1	iORR 24.5%	PFS 7 OS 14
	KAMILLA phase IIIb trial [30]	398/126	T-DM1	iORR 21.4% iCBR 43%	PFS 5.5 OS 18.9
	DESTINY–Breast 03 [37] phase III trial	39	T-DM1	iORR 33.4% iCR 2.8% iPR 30.6%	PFS 3
Trastuzumab deruxtecan (T-DXd)	DESTINY–Breast 03 [37] phase III trial	43	T-DXd	iORR 63.9% iCR 27.8% iPR 36.1%	PFS 15
	DESTINY–Breast 01 (Jerusalem [36]	24	T-DXd	iORR 58.3% iCR 4.2% iPR 54.2% iSD 33.3%	PFS 18
	DEBBRAH phase II trial [38]	39 HER2+ or HER2−low	T-DXd	iORR 44% iCBR 55%	-
Lapatinib	Lin 2009 phase II trial [39]	242	Lapatinib alone	iORR 6% Volumetric reduction >= 20–21%	PFS 2.4 OS 6.4
	Lin 2009 phase II trial [39]	50	Lapatinib plus capecitabine	iORR 20%	-
	Cetin [40] retrospective study	203/85	Lapatinib plus capecitabine	iORR 27% iCR 2.4% iPR 24.7%	PFS 7 OS 13
	Sutherland 2010 [41] prospective study	356/34	Lapatinib plus capecitabine	iORR 21%	PFS 4.5
	Petrelli 2017 [42] pooled analysis	799	Lapatinib plus capecitabine	iORR 29.2%	PFS 4.1 OS 12.2
	Lin 2011 [45] phase II trial	13 9	Lapatinib plus capecitabine Lapatinib + topotecan	iORR 38% iCBR 84% iORR 0%	-
	LANDSCAPE phase II trial [43]	45 before WBRT	Lapatinib plus capecitabine	iORR 65.9%	-
	LAPTEM phase I trial [44]	16	Lapatinib plus temozolomide	iORR 66.7%	PFS 2.6 OS 10.9
Pyrotinib	PERMEATE [54] single-arm phase II study	78 Cohort A (without RT) Cohort B (after RT)	Pyrotinib plus capecytabine	iORR 74.6% iORR 42%	PFS 11.3 PFS 5.6
Neratinib	TBCRC 022 phase II [46]	Arm 1: 40	Neratinib alone	iORR 8%	-
	TBCRC 022 phase II non randomized [46]	Arm 3A: 35	Neratinib plus capecitabine without previous lapatinib	iORR 49%	PFS 5.5 OS 13
	TBCRC 022 phase II [46]	Arm 3B:25	Neratinib plus capecitabine after lapatinib	iORR 33%	PFS 3 OS 15
Afatinib	LUX-Breast 2 phase II trial [49]	Arm A 40	Afatinib alone	iCBR 30%	-
	LUX-Breast 2 phase II trial [49]	Arm B 38	Afatinib plus vinorelbine	iCBR 34.2%	-
	LUX-Breast 2 phase II trial [49]	Arm C 43	Treatment of physician choice	iCBR 41.9%	-
Tucatinib	HER2CLIMB phase II trial [52]	291	Tucatinib plus trastuzumab plus capecitabine	iORR 47.3%	PFS 9.9 OS 21.6
Epertinib	Macpherson 2019 phase I/II trial [55]	45/5 Arm A Arm B Arm C	Epertinib plus trastuzumabTrastuzumab plus vinorelbineTrastuzumab plus capecitabine	iORR 67% iORR 0% iORR 50%	-
Bevacizumab	Lin 2013 phase II trial [57]	38	Bevacizumab plus carboplatin +/− trastuzumab	iORR 63%	-
Cabozantinib	Leone 2020 phase II trial [56]	Cohort 1: 21 HER2+ Cohort 2: 7 ER + HER2- Cohort 3: 8 TNBC	Cabozantinib +/− trastuzumab	iORR 5% iORR 14% iORR 0%	-
Everolimus	LCCC 1025 phase II trial [62]	32	Everolimus plus vinorelbine plus trastuzumab	iORR 4% iCBR3m. 65% iCBR6m. 27%	OS 12.2
Abemaciclib	JPBO [63] phase II trial	Cohort B	Abemaciclib plus trastuzumab	iORR 0% iCBR 11%	OS 10
	**Luminal breast cancer brain metastases**				
Abemaciclib	JPBO [63] phase II trial	Cohort A	Abemaciclib monotherapy or with endocrine therapy	iORR 5.2% iCBR 24%	OS 12.5
	**Triple-negative breast cancer brain metastases**				
PARP	TBCRC 018 phase II trial [64]	34	Iniparib plus irinotecan	iORR 12% iCBR 27%	TTP 2.1 OS 7.8
Ang1005	Kumthecar 2020 [66]	72	ANG1005 alone	iORR 8% iCBR 77%	OS 8
Eribulin	Adamo 2019 [68]	118	Eribulin mesylate	iORR 16%	PFS 5.5 OS 31.8
Sacituzumab govitecan	ASCENT phase III trial [69]	529/61	Sacituzumab govitecan	iORR 3% iCBR 9.4%	PFS 2.8 OS 6.8

* iORR—intracranial objective response rate (complete response (CR) + partial response (PR)); iCBR—intracranial clinical benefit rate—(CR + PR + SD (stable disease)); ER—estrogen receptor; OS—overall survival; PFS—progression-free survival; T-DM1—trastuzumab emtansine; T-DXd—trastuzumab deruxtecan; TNBC—triple negative breast cancer; TTP—time to progression.

**Table 2 cancers-14-00965-t002:** The most effective systemic drugs used in breast cancer patients with brain metastases (intracranial objective response rate iORR > 30%).

Main Agent	Scheme of the Treatment	iORR (CR + PR)
Pyrotinib [54]	Pyrotinib plus capecitabine before WBRT	74.6%
Lapatinib [43]	Lapatinib plus capecitabine before WBRT	65.9%
T-DXd [36,37]	T-DXd	58.3–63.9%
Neratinib [46]	Neratinib plus capecytabine without previous lapatinib	49%
Tucatinib [52]	Tucatinib plus trastuzumab plus capecitabine	47.3%
Neratinib [46]	Netatinib plus capecytabine after previous lapatinib	33%
T-DM1 [26,27,37]	T-DXd	21–44%
Afatinib [49]	Afatinib plus vinorelbine	34%

iORR—intracranial objective response rate (complete response (CR) + partial response (PR)); T-DM1—trastuzumab emtansine; T-DXd—trastuzumab deruxtecan; WBRT—whole-brain radiation therapy.

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
