# Peer review of "Intracranial Response Rate in Patients with Breast Cancer Brain Metastases after Systemic Therapy"

_cancers, 2022, doi:10.3390/cancers14040965_

Round 1

Reviewer 1 Report

The article "Intracranial response rate in patients with breast cancer brain metastases after systemic therapy" is a very interesting study in clinical terms. In this manuscript, Authors discussed the treatment of breast cancer patients with brain metastases based on new therapeutic regimens.
The article is properly prepared. After a short introduction, the Authors described the action of targeted drugs at the molecular level and their clinical effects. Synthetically developed tables are very important, taking into account the drugs the most effective in the systemic treatment of patients with breast cancer with brain metastases.
Conclusions are also well drafted. It is important to provide an explanation of the most frequently used abbreviations in the article.
The References section is based on the most recent manuscripts published in the last 10. years, including the most recent, i.e. from 2021.
Due to the very important clinical aspect and the summary of important information for oncologists on new targeted drugs used in the treatment of patients with breast cancer with brain metastases, I believe that the publication "Intracranial response rate in patients with breast cancer brain metastases after systemic therapy" should be accepted for printing in CANCERS in present form.

Author Response

Thank you very much for your recommendation and comments.

Point 1. It is important to provide an explanation of the most frequently used abbreviations in the article.

Response 1. The manuscript was checked and explanations of the abbreviations were provided if were missed.

Reviewer 2 Report

This review encompasses the current treatments of brain metastases with some interesting informations on recent advancements. The paper is easily readable. The use of discursive style instead of the insertion of plethoric table is appreciated

However the Authors could insert a brief discussion on investigational approaches.

Example line 53 introduction. Perivascular proliferation and activation of neoangiogenesis could be necessary for building a permeable brain-tumor barrier. It is known that neovascularization can occur in response to hypoxia in large metastases with central necrosis ( see Bailleux C et al. Br J Cancer 2021;124(1):142). This observation could be of interest since micrometastases and and leptomeningeal carcinomatosis are usually less vascularized and the blood-brain barrier less permeable than in macrometases. Therefore the therapeutic approach could be different.

Another observation  that is currently evaluated is that metastases from breast Cancer have a predilection for cerebellum. The different surface characteristics of endothelial cells in this site should be briefly reported ( Schroeder T et al. J Neuroncol 2020; 147(1):229)

Line 335 : nivolumab and not niwolumab

Please insert some words on multimodality approach ( combination of immunotherapy with Cancer vaccines or stereotactic RT ( example NCT03807765,

Author Response

Thank you very much for your comments.

Point 1.  The Authors could insert a brief discussion on investigational approaches.

Example line 53 introduction. Perivascular proliferation and activation of neoangiogenesis could be necessary for building a permeable brain-tumor barrier. It is known that neovascularization can occur in response to hypoxia in large metastases with central necrosis ( see Bailleux C et al. Br J Cancer 2021;124(1):142). This observation could be of interest since micrometastases and and leptomeningeal carcinomatosis are usually less vascularized and the blood-brain barrier less permeable than in macrometases. Therefore the therapeutic approach could be different.

Another observation  that is currently evaluated is that metastases from breast Cancer have a predilection for cerebellum. The different surface characteristics of endothelial cells in this site should be briefly reported ( Schroeder T et al. J Neuroncol 2020; 147(1):229)

Response 1. Thank you for recommendation. We have added and discussed the proposed data in Introduction section - the new references 18. and 19. in the manuscript after changes. We agree that extended information can be of interest for readers.

Point 2. Line 335 : nivolumab and not niwolumab

Response 2. It was corrected for “nivolumab”.

Point 3. Please insert some words on multimodality approach ( combination of immunotherapy with Cancer vaccines or stereotactic RT ( example NCT03807765,

Response 3. The ongoing trials which assess radiation therapy combined with immune therapy were indicated in “Immune therapy” section including mentioned NCT03807765 trial (new reference 72).  We added some information about cancer vaccines assessed with immune therapy - also in “Immune therapy” section.

Reviewer 3 Report

The review provides an up-to-date analysis of available and emerging treatment protocols for patients with breast cancers brain metastases. The authors present results of multiple clinical trials involving patients with HER2 positive tumors, luminal, and triple-negative breast cancers. Even though there are multiple reviews on that subject available, the manuscript could be  a valuable tool for oncologists treating patients with breast cancer CNS metastases

As a minor comment, authors should update the paragraphs' enumeration to better recapitulate the logic flow of the review.:

  1. HER2-positive Breast Cancer

3.1 Monoclonal Antibodies

3.2 TKIs

3.3 Other Targeted Therapies

  1. Luminal Breast Cancer

5 TNBC

Page 7, line 251 remove “it” after lapatinib

Author Response

Thank you very much for your comments.

Point 1. As a minor comment, authors should update the paragraphs' enumeration to better recapitulate the logic flow of the review.:

  1. HER2-positive Breast Cancer

3.1 Monoclonal Antibodies

3.2 TKIs

3.3 Other Targeted Therapies

  1. Luminal Breast Cancer

5 TNBC

Response 1. We fully agree with comment. We changed the numeration in manuscript and added “Antibody-drug conjugates” section.

Point 2.  Page 7, line 251 remove “it” after lapatinib

Response 2. Thank you for proposition – we deleted “it”.

Reviewer 4 Report

The authors have done an extensive and complete review of intracranial responses with available or investigational systemic therapies in metastatic breast cancer. The different tables in the paper are clear and useful for the reader.

However several recent reviews dealing with the same subject are already published (eg: Bailleux, C et al. Br J Cancer 124, 142–155 (2021); Corti C et al Cancer Treat Rev. 2022 Feb;103:102324. Can the authors argue what is the added value of this review as compared to the existing ones ? The presentation of the review and the general organization should be modified and highlight the originality of the review if any.

The English in the present manuscript requires improvement.

Author Response

Thank you very much for your comments.

Point 1. Several recent reviews dealing with the same subject are already published (eg: Bailleux, C et al. Br J Cancer 124, 142–155 (2021); Corti C et al Cancer Treat Rev. 2022 Feb;103:102324. Can the authors argue what is the added value of this review as compared to the existing ones ? The presentation of the review and the general organization should be modified and highlight the originality of the review if any.

Response 1. Our review summarizes the current knowledge on this topic, including very hot results of clinical trials which were presented during San Antonio Breast Cancer Symposium 2021 (new DEBBRAH study and subgroup analysis of DESTINY-Breast03 and HER2CLIMB trials concerning brain metastases patients - the references 37., 38. and 52. in the manuscript after changes). There is no publication discussing these latest data including those mentioned by Reviewer. We submitted our review to Cancers 2 weeks after SABCS 2021. We believe that updated data are important for oncologists. We indicated in the Summary, the Abstract and now highlighted in “Aim” section that we also analyzed very recent advancements in the field.  

Point 2. The English in the present manuscript requires improvement.

Response 2. The manuscript was checked by native speaker.

Reviewer 5 Report

Dear authors, well written review article on "Intracranial response rate in patients with breast cancer brain metastases after systemic therapy"

I only have a minor remark: reference 9 ist not cited correctly (Clin Trial)

Yours sincerely

Author Response

Thank you very much for your recommendation and comment.

Point 1. A minor remark: reference 9 ist not cited correctly (Clin Trial)

Response 1. We checked the cited publication. It is a review which summarizes some modalities used in breast cancer patients with brain metastases. We think it is cited correctly. In that sentence we did not want to put information about the strict clinical trial, rather clinical practice and information about the number of ongoing trials which was presented in cited article.